# The complete mitogenome of *Lysmata vittata* (Crustacea: Decapoda: Hippolytidae) with implication of phylogenomics and population genetics

**Longqiang Zhu**[1,2,3], **Zhihuang Zhu**[1,2]*, **Leiyu Zhu**[1,2], **Dingquan Wang**[3], **Jianxin Wang**[3]*, **Qi Lin**[1,2,3]*

**1** Fisheries Research Institute of Fujian, Xiamen, China, **2** Key Laboratory of Cultivation and High-value Utilization of Marine Organisms in Fujian Province, Xiamen, China, **3** Marine Microorganism Ecological & Application Lab, Zhejiang Ocean University, Zhejiang, China

* xmqlin@sina.com (QL); jxwang@zjou.edu.cn (JW); zhu.zhi.huang@163.com (ZZ)

## Abstract

In this study, the complete mitogenome of *Lysmata vittata* (Crustacea: Decapoda: Hippolytidae) has been determined. The genome sequence was 22003 base pairs (bp) and it included thirteen protein-coding genes (PCGs), twenty-two transfer RNA genes (tRNAs), two ribosomal RNA genes (rRNAs) and three putative control regions (CRs). The nucleotide composition of AT was 71.50%, with a slightly negative AT skewness (-0.04). Usually the standard start codon of the PCGs was ATN, while *cox1*, *nad4L* and *cox3* began with TTG, TTG and GTG. The canonical termination codon was TAA, while *nad5* and *nad4* ended with incomplete stop codon T, and *cox1* ended with TAG. The mitochondrial gene arrangement of eight species of the Hippolytidae were compared with the order of genes of Decapoda ancestors, finding that the gene arrangement order of the *Lebbeus groenlandicus* had not changed, but the gene arrangement order of other species changed to varying degrees. The positions of the two tRNAs genes (*trnA* and *trnR*) of the *L. vittata* had translocations, which also showed that the Hippolytidae species were relatively unconserved in evolution. Phylogenetic analysis of 50 shrimp showed that *L. vittata* formed a monophyletic clade with *Lysmata*/*Exhippolysmata* species. This study should be helpful to better understand the evolutionary status, and population genetic diversity of *L. vittata* and related species.

## Introduction

The genus *Lysmata* is an important group in family Hippolytidae, contains more than 48 described species, most of which are small shrimp living in shallow waters [1,2]. For a long time, the classification of Hippolytidae was the most controversial family in Decapoda, especially the monophyly of Hippolytidae and the position of the genus *Lysmata* [3,4]. In the past few decades, the studies of *Lysmata* mainly focused on morphology, with relatively few studies on population genetic structure. Meanwhile, most of the selected marker genes are partial

**Data Availability Statement:** Data are available from the NCBI database (accession number MT478132).

**Funding:** This study was supported by the special fund of marine and Fishery Structure Adjustment in Fujian (No.2017HYJG03, No.2020HYJG01, No.2020HYJG08), the National key R&D Program of China (2019YFD0901305), the Science and Technology Program of Zhoushan (2019C21011), the Natural Science Foundation of Zhejiang Province, China (LY12C03003) and the Province Key Research and Development Program of Zhejiang (2021C02047).

**Competing interests:** The authors declare there are no competing interests.

sequences of *rrnL*, *rrnS* and *cox1*, and these gene fragments often fail to provide enough information to make the study of population genetics and species evolution.

The mitogenome is a significant tool for studying identification and phylogenetic relationships in the different species [5]. In shrimps, the mitochondria is maternally inherited, usually is circular and approximately 15 to 20 kb in length, including thirteen PCGs, two rRNAs, twenty-two tRNAs and one CR. The mitogenome contains abundant gene information, and the phylogenetic tree based on the mitogenome sequences has the advantages of stable and reliable structure. Analyzing the genetic relationship of species through the establishment of the 13PCGs sequence of the mitogenome can better solve the problems encountered in species classification.

*Lysmata vittata* (Crustacea: Decapoda: Hippolytidae) belongs to a small marine ornamental shrimp, commonly known as peppermint shrimp, which is popular in the marine aquarium trade. The species has a special sexual system, ie, protandric simultaneous hermaphrodite (PSH) [3]. It is a member of the clean shrimp family, a common marine ornamental species that originated in the Indian Ocean-Pacific region, including coastal areas such as China, Japan, Philippines and Australia [6–8]. *L. vittata* prefers to move in the range of 2~50 m below the sea surface, usually hiding in the reef during the day and activating at night [9]. In recent years, with the continuous breakthroughs in genomics technology, the phylogenetic research of the *Lysmata* species has gradually moved from the morphological level to the genome level. As a relatively important marine ornamental species, the determination of *L. vittata* mitogenome is of great significance for the development of genetic diversity and evolutionary history of *Lysmata*.

In this study, the mitogenome of the *L. vittata* has been successfully determined, and its structure and phylogenetic status have been analyzed. This work should help to further understand the evolutionary relationship and population genetic diversity between the *L. vittata* and related species.

## Materials and methods

### Mitochondria DNA sequencing and genome assembly

Specimens of *L. vittata* were collected in Xiamen, Fujian province, China. The morphological characteristics of the species follow the previous description of Abdelsalam [1]. Approximately 5g of muscle tissue was harvested for mtDNA isolation using an improved extraction method [10]. After DNA isolation, the isolated DNA was purified according to manufacturer's instructions (Illumina), and then 1 μg was taken to create short-insert libraries, whose insertion size was 430 bp, followed by sequencing on the Illumina Hiseq 4000 [11] (Shanghai BIOZERON Co., Ltd). The high molecular weight DNA was purified and used for PacBio library prep, BluePippin size selection, then sequenced on the Sequel Squencer.

The raw data obtained by sequencing was processed and then the duplicated sequences were assembled. The mitogenome was reconstructed using a combination of the PacBio Sequel and the Illumina Hiseq data. Assemble the genome framework by the both Illumina and PacBio using SOAPdenovo2.04 [12]. Verifying the assembly and completing the circle or linear characteristic of the mitogenome, filling gaps if there were. Finally, the clean data were mapped to the assembled draft mitogenome to correct the wrong bases, and the most of the gaps were filled through local assembly.

### Validation of mitogenome data

In order to ensure the accuracy of the *L. vittata* mitogenome data, we resequenced the samples on the Illumina HiSeq X10 platform (Nanjing Genepioneer Biotechnologies Co. Ltd).

## Genome annotation and sequence analysis

Mitogenome sequences were annotated using homology-based prediction and de novo prediction, and the EVidenceModeler v1.1 [13] was used to integrate the complete genetic structure. Twenty-two tRNAs and two rRNAs were predicted by tRNAscan-SE [14] and rRNAmmer 1.2 [15]. The circular of the complete *L. vittata* mitogenome graphical map was drawn using OrganellarGenomeDRAW v1.2 [16]. The RSCU of thirteen PCGs (remove incomplete codons) was calculated using MEGA 5.0 [17]. The composition skewness of each component of the genome was calculated according to the following formulas: AT-skew = (A-T)/(A+T); GC-skew = (G-C)/(G+C) [18]. The secondary cloverleaf structure of tRNAs was examined with MITOS WebServer (http://mitos2.bioinf.uni-leipzig.de/index.py) [19].

## Phylogenetic analysis

To reconstruct the phylogenetic relationship among shrimp, the PCGs sequences of the 49 Decapoda species were downloaded from GenBank database (S1 Table). The PCGs sequences of *Harpiosquilla harpax* (NC_006916) were used as outgroup. The nucleotide and amino acid sequences of 13 PCGs were aligned using MEGA 5.0 [17]. Gblocks was used to identify and selected the conserved regions [20]. Subsequently, Bayesian inference (BI) and Maximum likelihood (ML) analysis were utilized for reconstructing phylogenetic tree by MrBayes v3.2.6 [21] and PhyML 3.1 [22]. According to the Akaike Information Criterion (AIC) [23], TVM + I + G model was considered as the best-fit model for analysis with nucleotide alignments using jModeltest [24], and MtArt + I + G + F model was the optimal model for the amino acid sequence dataset using ProtTest 3.4.2 [25]. In BI analysis, two simultaneous runs of 10000000 generations were conducted for the matrix. Sampling trees every 1000 generations, and diagnostics were calculated every 5000 generations, with three heated and one cold chains to encourage swapping among the Markov-chain Monte Carlo (MCMC) chains. Additionally, the standard deviation of split frequencies was below 0.01 after 10000000 generations, and the potential scale reduction factor (PSRF) was close to 1.0 for all parameters. Posterior probabilities over 0.9 or bootstrap percentage over 75%, the results were regarded as credible [26,27]. The resulting phylogenetic trees were visualized in Fig Tree v1.4.0.

## Results and discussion

### Genome structure, organization and composition

The mitogenome of *L. vittata* was a typical circular molecule of 22003 bp in size. It contained 37 mitochondrial genes (thirteen PCGs, twenty-two tRNAs, two rRNAs and three CRs) (Fig 1 and S2 Table). Among the 37 genes, the coding direction of the twenty-three genes was clockwise (F-strand), and the coding direction of the remaining fourteen genes was counterclockwise (R-strand) (Fig 1 and S2 Table).

The nucleotide composition of the mitogenome was biased toward A and T (T = 37.15%, A = 34.35%, C = 16.69%, G = 11.80%) (Table 1). The relatively AT contents of the complete mitogenome were calculated [mitogenome (71.50%), PCGs (69.79%), tRNAs (69.58%) and rRNAs (69.29%)] (Tables 1 and 2). However, with the exception of *Thor amboinensis* (73.10%), the AT content of *L. vittata* mitogenome was higher than other species in the Hippolytidae (Table 1). Among the nine species of Hippolytidae, the AT-skew values of *L. vittata* (-0.039) was similar with *L. boggessi* (-0.040), and the AT-skew values of *Lebbeus groenlandicus* (0.062), *Exhippolysmata ensirostris* (0.009) and *Saron marmoratus* (0.110) was positive. In addition, with the exception of *Thor amboinensis* (-0.081), the GC-skew value for *L. vittata* (Guangdong) was the biggest negative comparing to that of other mitogenomes (Table 1). By

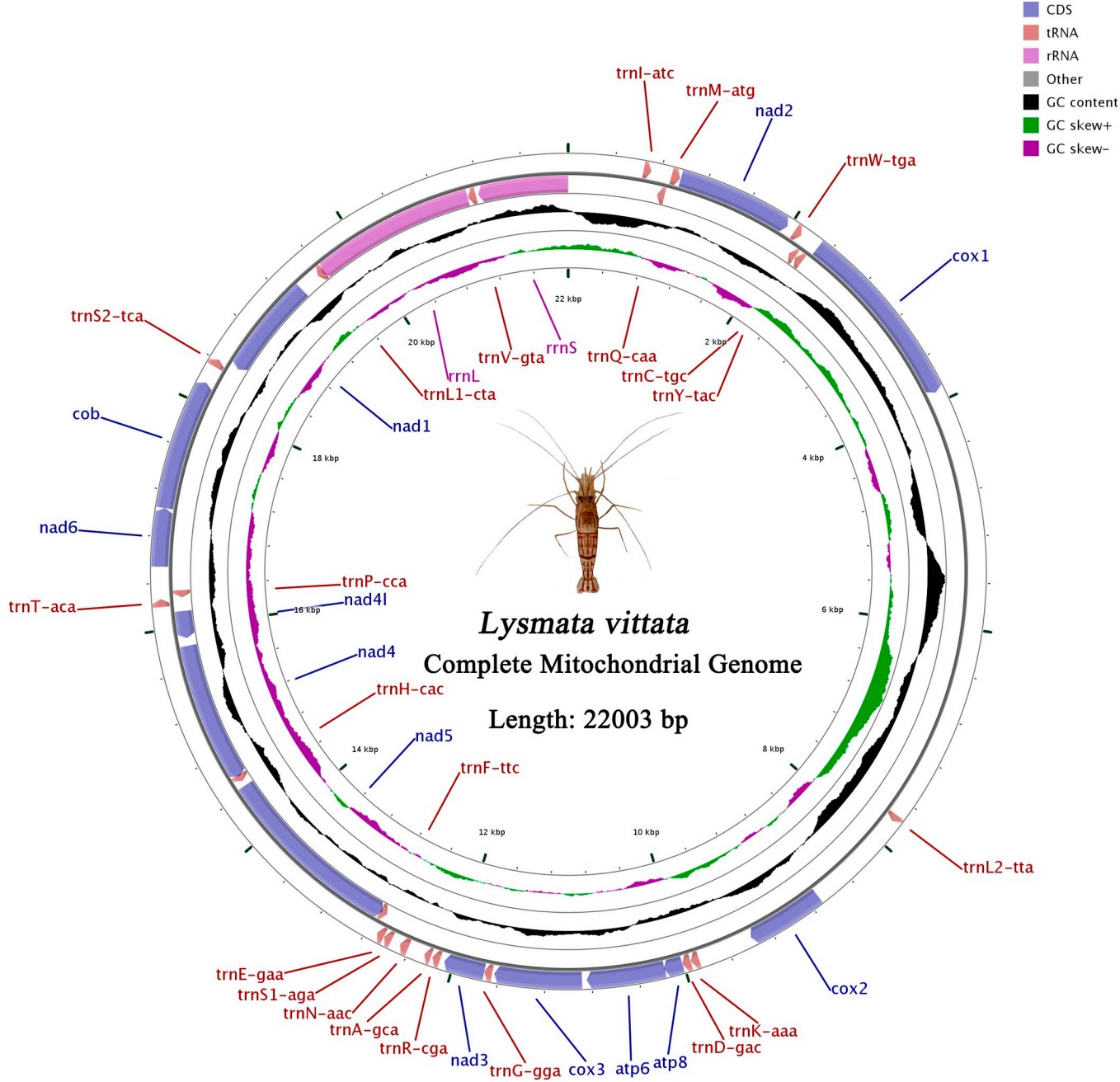

**Fig 1. Mitogenome map of *Lysmata vittata*.** The genes outside the map were coded on the F strand, whereas the genes on the inside of the map are coded on the R strand. The middle black circle displays the GC content and the inside purple and green circle displays the GC skew.

comparing the mitogenome sequence of *L. vittata* (Fujian) with that of *L. vittata* (Guangdong), it was found that the whole mitogenome sequence of *L. vittata* (Fujian) could completely overlap with *L. vittata* (Guangdong) except that it was 1146 bp bases longer than that of *L. vittata* (Guangdong). The base distribution of *L. vittata* (Guangdong) deletion was shown in S1 Fig. The reason for sequence deletion may be related to sequencing method and sequence splicing. All original sequence data in this study were submitted to the NCBI database under accession number MT478132.

**Table 1. Composition and skewness of mitogenome in 9 Hippolytidae species.**

| Species | Size (bp) | T% | C% | A% | G% | A+T % | ATskewness | GCskewness |
|---|---|---|---|---|---|---|---|---|
| **Whole genome** | | | | | | | | |
| ***L. vittata*** | **22003** | **37.15** | **16.69** | **34.35** | **11.80** | **71.50** | **-0.039** | **-0.172** |
| *L. vittata* (Guangdong) | 20857 | 36.87 | 17.03 | 34.36 | 11.74 | 71.23 | -0.035 | -0.184 |
| *L. amboinensis* | 16735 | 32.36 | 21.65 | 31.68 | 14.31 | 64.05 | -0.011 | -0.204 |
| *L. debelius* | 16757 | 34.11 | 19.70 | 33.04 | 13.15 | 67.15 | -0.016 | -0.199 |
| *L. boggessi* | 16979 | 35.01 | 19.55 | 32.33 | 13.10 | 67.34 | -0.040 | -0.197 |
| *L. groenlandicus* | 17398 | 30.37 | 21.37 | 34.41 | 13.85 | 64.78 | 0.062 | -0.213 |
| *E. ensirostris* | 16350 | 31.93 | 21.31 | 32.52 | 14.24 | 64.45 | 0.009 | -0.199 |
| *S. marmoratus* | 16330 | 30.21 | 21.70 | 37.68 | 10.42 | 67.89 | 0.110 | -0.351 |
| *T. amboinensis* | 15553 | 37.01 | 14.54 | 36.09 | 12.36 | 73.10 | -0.013 | -0.081 |
| **PCGs** | | | | | | | | |
| ***L. vittata*** | **11144** | **41.09** | **15.25** | **28.70** | **14.96** | **69.79** | **-0.178** | **-0.010** |
| *L. vittata* (Guangdong) | 11285 | 41.17 | 15.18 | 28.75 | 14.90 | 69.92 | -0.178 | -0.009 |
| *L. amboinensis* | 11192 | 36.66 | 19.12 | 25.94 | 18.28 | 62.60 | -0.171 | -0.022 |
| *L. debelius* | 11162 | 38.66 | 17.17 | 27.40 | 16.78 | 66.05 | -0.170 | -0.011 |
| *L. boggessi* | 11165 | 38.83 | 17.65 | 27.02 | 16.50 | 65.85 | -0.179 | -0.034 |
| *L. groenlandicus* | 11175 | 37.19 | 18.93 | 25.66 | 18.23 | 62.85 | -0.184 | -0.019 |
| *E. ensirostris* | 11062 | 36.77 | 19.30 | 26.05 | 17.87 | 62.83 | -0.171 | -0.038 |
| *S. marmoratus* | 11135 | 37.53 | 17.45 | 28.50 | 16.52 | 66.03 | -0.137 | -0.027 |
| *T. amboinensis* | 11178 | 41.39 | 13.36 | 30.26 | 14.99 | 71.65 | -0.155 | 0.058 |

## PCGs and codon usage

The PCGs region was 11144 bp long, and accounted 50.6% of the *L. vittata* mitogenome. Furthermore, a contrast of nucleotide composition, AT-skew value, and GC-skew value of PCGs

**Table 2. Composition and skewness of *Lysmata vittata* mitogenome.**

| *Lysmata vittata* | Size(bp) | T (%) | C (%) | A (%) | G (%) | A+T (%) | AT-skew | GC-skew |
|---|---|---|---|---|---|---|---|---|
| atp6 | 675 | 40.15 | 19.41 | 28.30 | 12.15 | 68.44 | -0.17 | -0.23 |
| atp8 | 165 | 43.64 | 15.76 | 35.15 | 5.45 | 78.79 | -0.11 | -0.49 |
| cob | 1137 | 39.40 | 20.14 | 27.88 | 12.58 | 67.28 | -0.17 | -0.23 |
| cox1 | 1614 | 37.73 | 17.91 | 27.76 | 16.60 | 65.49 | -0.15 | -0.04 |
| cox2 | 693 | 37.95 | 19.77 | 28.43 | 13.85 | 66.38 | -0.14 | -0.18 |
| cox3 | 756 | 39.29 | 18.25 | 27.91 | 14.55 | 67.20 | -0.17 | -0.11 |
| nad1 | 927 | 44.01 | 10.79 | 27.29 | 17.91 | 71.31 | -0.23 | 0.25 |
| nad2 | 1005 | 43.28 | 18.01 | 29.05 | 9.65 | 72.34 | -0.20 | -0.30 |
| nad3 | 354 | 42.66 | 18.93 | 26.27 | 12.15 | 68.93 | -0.24 | -0.22 |
| nad4 | 1336 | 43.11 | 9.51 | 28.59 | 18.79 | 71.70 | -0.20 | 0.33 |
| nad4l | 246 | 45.12 | 7.72 | 26.02 | 21.14 | 71.14 | -0.27 | 0.46 |
| nad5 | 1732 | 41.17 | 9.82 | 31.64 | 17.38 | 72.81 | -0.13 | 0.26 |
| nad6 | 504 | 44.64 | 17.06 | 28.57 | 9.72 | 73.21 | -0.22 | -0.27 |
| tRNAs | 1512 | 33.27 | 14.02 | 36.31 | 16.40 | 69.58 | 0.04 | 0.08 |
| rRNAs | 2315 | 32.40 | 11.88 | 36.89 | 18.83 | 69.29 | 0.06 | 0.23 |
| CR1 | 650 | 42.15 | 9.85 | 38.31 | 9.69 | 80.46 | -0.05 | -0.01 |
| CR2 | 3821 | 38.50 | 14.39 | 33.73 | 13.37 | 72.23 | -0.07 | -0.04 |
| CR3 | 888 | 42.34 | 13.51 | 34.91 | 9.23 | 77.25 | -0.10 | -0.19 |

from other species in the Hippolytidae were also exhibited in Table 1. Nine of thirteen PCGs (*atp6*, *atp8*, *cob*, *cox1-3*, *nad2-3* and *nad6*) were encoded on the light (F) strand, while the other four genes (*nad1*, *nad4L* and *nad4-5*) were encoded on the heavy (R) strand (S2 Table). Each PCG was initiated by a canonical ATN codon (ATG for *atp6*, *atp8*, *nad2-5* and *cob*; ATT for *cox2 and nad1*; ATC for *nad6*), except for *cox1* (TTG), *nad4L* (TTG) and *cox3* (GTG) (S2 Table). Two of the thirteen PCGs (*nad5* and *nad4*) terminated with incomplete stop codon T, one PCG (*cox1*) terminated with stop codon TAG, and the other ten PCGs terminated with the canonical termination codon TAA (S2 Table).

The RSCU values of *L. vittata* mitogenome were analyzed and the results were shown in Table 3. The total number of codons in thirteen PCGs was 3714 except eleven canonical stop codons and two incomplete stop codons and the most common amino acids were Ile (AUR) (499), Phe (UUR) (357) and Leu2 (UUR) (315), whereas codons encoding Cys (UGR) (41) and Met (AUR) (24) were rare (Fig 2). The overall A + T content of thirteen PCGs was 69.79%, the AT-skews and GC-skews were negative which implied a higher occurrence of Ts and Cs than As and Gs (Table 1).

## Transfer RNAs and ribosomal RNAs

The mitogenome of *L. vittata* contained twenty-two tRNAs and these genes ranged from 60 (*trnA*) to 77 bp (*trnN*) (S2 Table). The tRNAs showed a strong A +T bias (69.58%), while they also exhibited positive AT-skew (0.04) and GC-skew (0.08) (Table 1). Eight tRNAs [*trnQ* (CAA), *trnC* (UGC), *trnY* (UAC), *trnF* (UUC), *trnH* (CAC), *trnP* (CCA), *trnL1* (CUA) and *trnV* (GUA)] were present on the R strand and the remaining fourteen were present on the F strand (S2 Table). The examined secondary structure of twenty-two tRNAs was shown in S2 Fig. The other twenty-one tRNAs had typical cloverleaf secondary structure except that *trnS1* (AGA) lacked the dihydropyridine (DHU) arm [18,19,27,28] (S1 Fig). In the secondary structure of the tRNAs, the most common non-Watson–Crick base pair was G–U (e.g. *trnC*, *trnE*), followed by U–U (e.g. *trnA*, *trnC*) [19]. In addition, several mismatches were common in tRNAs, such as A–C (e.g. *trnA*), C–U (e.g. *trnA*, *trnG*) and A–A (e.g. *trnM*, *trnS1*) (S1 Fig).

**Table 3. The codon number and relative synonymous codon usage (RSCU) in *L. vittata* mitochondrial protein coding genes.**

| Codon | Count | RSCU | Codon | Count | RSCU | Codon | Count | RSCU | Codon | Count | RSCU |
|---|---|---|---|---|---|---|---|---|---|---|---|
| UUU(F) | 300 | 1.68 | UCU(S) | 129 | 2.46 | UAU(Y) | 101 | 1.57 | UGU(C) | 32 | 1.56 |
| UUC(F) | 57 | 0.32 | UCC(S) | 29 | 0.55 | UAC(Y) | 28 | 0.43 | UGC(C) | 9 | 0.44 |
| UUA(L) | 283 | 3.13 | UCA(S) | 92 | 1.76 | UAA(*) | 10 | 0.29 | UGA(W) | 92 | 2.68 |
| UUG(L) | 32 | 0.35 | UCG(S) | 12 | 0.23 | UAG(*) | 1 | 0.03 | UGG(W) | 15 | 1 |
| CUU(L) | 131 | 1.45 | CCU(P) | 101 | 2.71 | CAU(H) | 53 | 1.47 | CGU(R) | 12 | 0.4 |
| CUC(L) | 33 | 0.36 | CCC(P) | 14 | 0.38 | CAC(H) | 19 | 0.53 | CGC(R) | 2 | 0.07 |
| CUA(L) | 59 | 0.65 | CCA(P) | 28 | 0.75 | CAA(Q) | 55 | 1.62 | CGA(R) | 38 | 1.26 |
| CUG(L) | 5 | 0.06 | CCG(P) | 6 | 0.16 | CAG(Q) | 13 | 0.38 | CGG(R) | 11 | 0.36 |
| AUU(I) | 266 | 1.6 | ACU(T) | 85 | 1.95 | AAU(N) | 108 | 1.65 | AGU(S) | 45 | 0.86 |
| AUC(I) | 42 | 0.25 | ACC(T) | 23 | 0.53 | AAC(N) | 23 | 0.35 | AGC(S) | 7 | 0.13 |
| AUA(I) | 191 | 1.15 | ACA(T) | 61 | 1.40 | AAA(K) | 83 | 1.77 | AGA(S) | 93 | 3.08 |
| AUG(M) | 24 | 1 | ACG(T) | 5 | 0.11 | AAG(K) | 11 | 0.23 | AGG(S) | 25 | 0.83 |
| GUU(V) | 95 | 1.82 | GCU(A) | 93 | 2.14 | GAU(D) | 56 | 1.51 | GGU(G) | 61 | 1.06 |
| GUC(V) | 8 | 0.15 | GCC(A) | 25 | 0.57 | GAC(D) | 18 | 0.49 | GGC(G) | 20 | 0.35 |
| GUA(V) | 87 | 1.67 | GCA(A) | 50 | 1.15 | GAA(E) | 64 | 1.35 | GGA(G) | 106 | 1.84 |
| GUG(V) | 19 | 0.36 | GCG(A) | 6 | 0.14 | GAG(E) | 31 | 0.65 | GGG(G) | 43 | 0.75 |

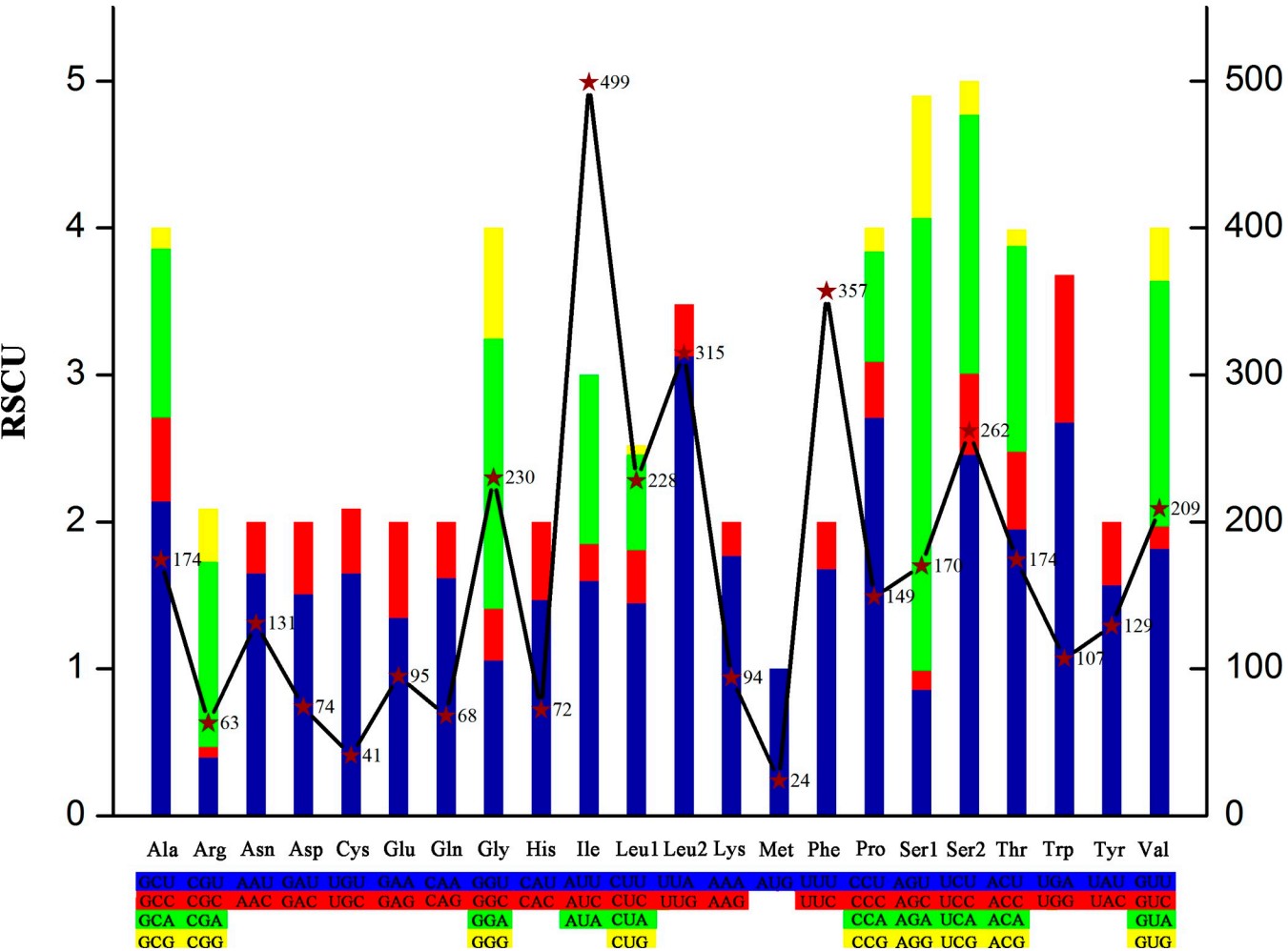

**Fig 2. RSCU and Codon distribution in the mitogenome of *L. vittata*.** The left ordinate represents RSCU, and the right ordinate represents the number of the Codon distribution.

Two rRNA genes were found on the R strand. The *rrnL* was 1494 bp and *rrnS* was 821 bp, one located between *trnL1* and *trnV* and another located between *trnV* and CR1 (S2 Table and Fig 1). The total A+T content of the two rRNAs was 69.29%, with a positive AT-skew (0.06) (Table 2).

## Overlapping and intergenic regions

The mitogenome of *L. vittata* contained four overlapping regions, these four pairs of genes were presented: *atp8/atp6*, *trnE/trnF*, *nad4/nad4L* and *trnL1/rrnL*, with the longest 23 bp overlap located between *trnL1* and *rrnL* (S2 Table). The 27 intergenic regions were found with a length varying from 2 ~ 3821 bp (S2 Table). Three putative CRs had been identified in *L. vittata* mitogenome. The CR1 was located between *rrnS* and *trnI*, with a length of 650 bp, and the A+T content was 80.46%. The CR2 was located between *cox1* and *trnL2*, with a length of 3821 bp, and the A+T content was 72.23%. The CR3 was located between *trnL2* and *cox2*, with a length of 888 bp, and the A+T content was 77.25% (Tables 2 and S2).

To our knowledge, the complete mitogenome sequence of *L. vittata* is the longest in the existing research on shrimp. How multiple CRs were generated and evolved in the

mitogenome of *Lysmata* is a novel problem that has not yet been solved, and more mitogenomes of *Lysmata* are still needed to clarify the mechanism forming this phenomenon.

## Gene rearrangement

In terms of gene rearrangement, compared with the genes order of the ancestor of Decapoda [20,29], the order of the genes of *L. groenlandicus* remains unchanged, and all species of the *Lysmata* had multiple CR regions. Among them, *L. amboinensis*, *L. debelius* and *L. boggessi* had 2 CR regions, *L. vittata* has three CR regions and the positions of the two tRNA genes (*trnA* and *trnR*) had been translocated. In addition, the mitochondrial genes order of *E. ensirostris*, *S. marmoratus* and *T. amboinensis* all had varying degrees of translocation compared

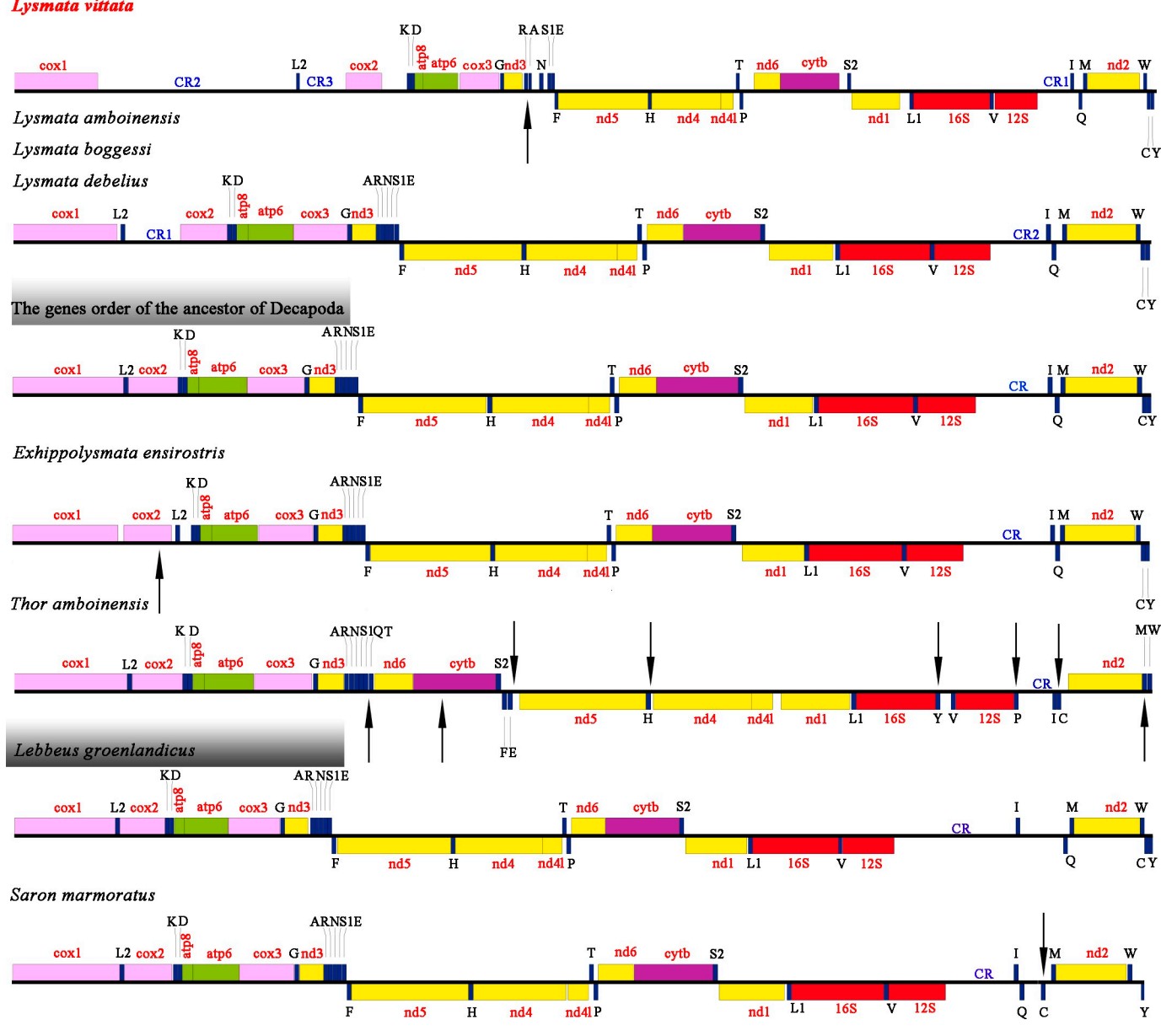

**Fig 3. Linear representation of gene rearrangements of Hippolytidae species.**

with the gene order of Decapoda (Fig 3). The position of *cox2* and *trnL2* of *E. ensirostris* was translocated, and the *trnC* and gene block (*trnM-nad2-trnW*) of *S. marmoratus* were translocated. *T. amboinensis* produced more translocations, including two gene block (*nad6-cob-trnS2 and nad5-trnH-nad4-nad4l*) translocations and single tRNA (*trnQ, trnT, trnE, trnH, trnY, trnP, trnC and trnM*) translocations (Fig 3). In fact, gene rearrangement was a very common phenomenon in the mitogenome and the rearrangement mainly occurred in tRNA genes. Gene arrangement was stable, and it could be used as an important phylogenetic marker in the analysis of evolutionary perspective on shrimp. Comparing the order of the

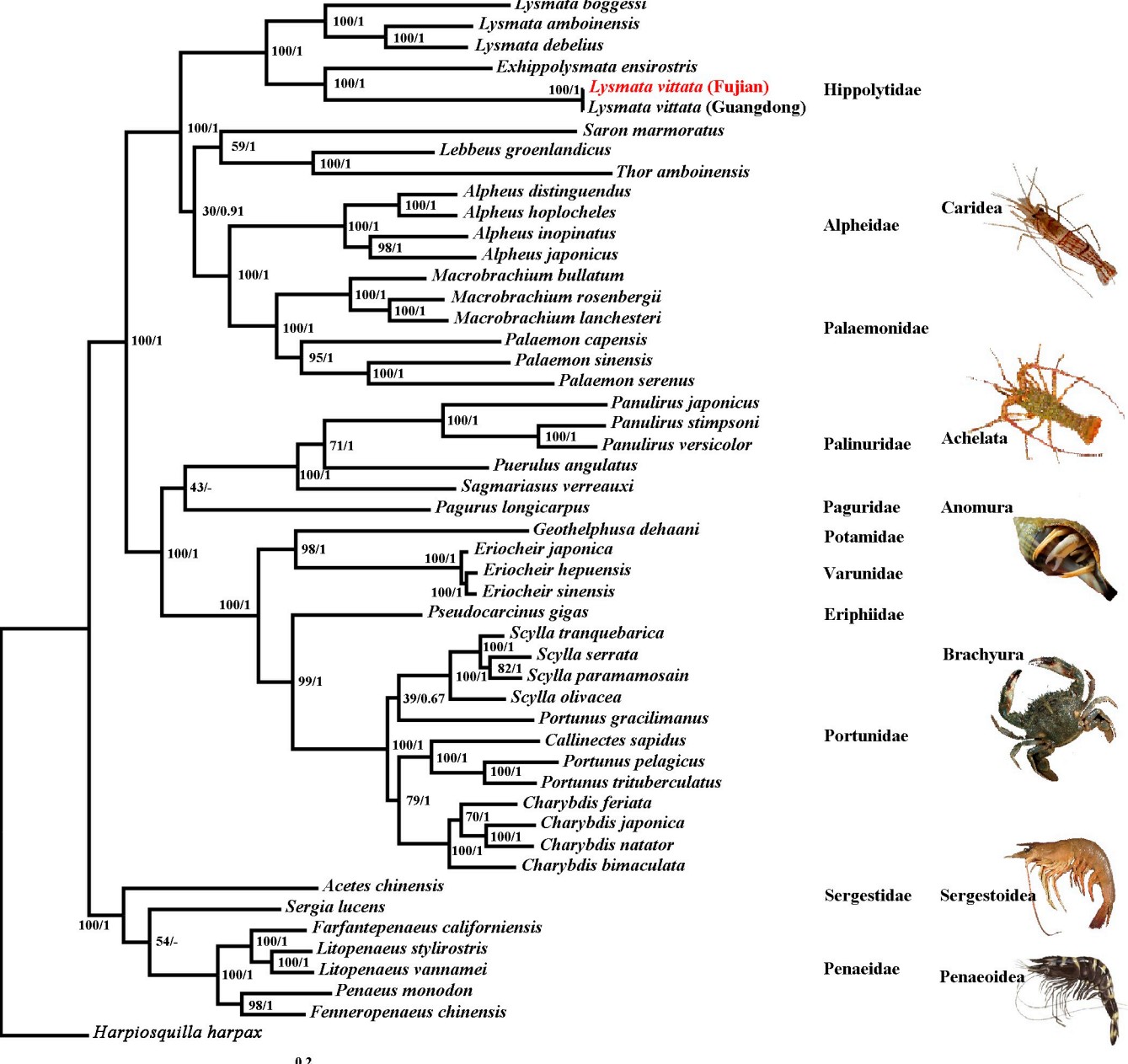

**Fig 4. Phylogenetic tree inferred from nucleotide sequences of 13 PCGs of the mitogenome using ML and BI methods (BP/PP).**

mitochondrial genes of various species of Hippolytidae, it indicates that the species of Hippolytidae are not conserved in evolution.

## Phylogenetic analysis

The taxonomic status of genus *Lysmata* within Hippolytidae has been a highly contentious issue for a long time. In this study, using ML and BI analysis methods, phylogenetic analysis was performed based on the nucleotide and amino acid sequences of thirteen PCGs of the species in S1 Table, and the analysis results were presented (Figs 4 and 5). The phylogenetic tree based on the nucleotide sequence of thirteen PCGs showed that *Lysmata* and *Exhippolysmata* formed a monophyletic group, while *S. marmoratus*, *L. groenlandicus* and *T. amboinensis* was clustered into a monophyletic group with species of Alpheidae and Palaemonidae (Fig 4). This analysis supported Christoffersen's [30,31] proposal to classify the *Lysmata* into the same classification level as the Lysmatidae. The phylogenetic tree based on the amino acid sequence of 13 PCGs revealed that the species of Hippolytidae clustered into a large branch, among which *Lysmata-Exhippolysmata* formed a monophyletic branch, which was in sister relationship with *S. marmoratus-L. groenlandicus/T. amboinensis* (Fig 5). The topological structures of the

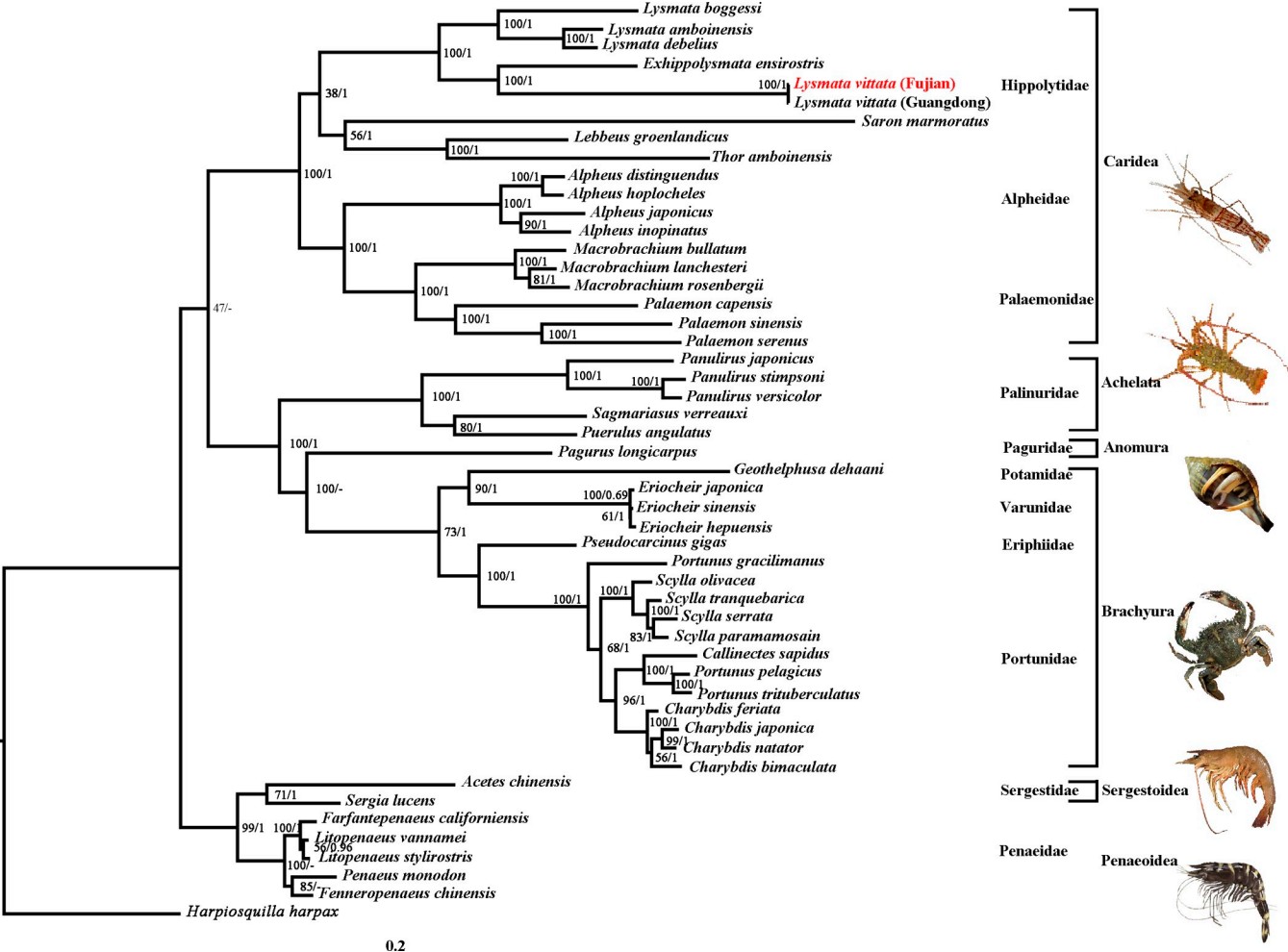

**Fig 5. Phylogenetic tree inferred from amino acid sequences of 13 PCGs of the mitogenome using ML and BI methods (BP/PP).**

phylogenetic trees constructed based on the nucleotide sequence and amino acid sequence were slightly different within the Hippolytidae, but the monophyleticity of *Lysmata-Exhippolysmata* had been fully verified in previous studies [32–35]. Furthermore, the phylogenetic analyses confirmed that *L. vittata* (Fujian) and *L. vittata* (Guangdong) were closely related. The two shrimps were clustered together and the branch length was zero. Especially their branch nodes were strongly supported (ML BP = 100%; BI PP = 1), indicating that there was almost no difference between *L. vittata* (Fujian) and *L. vittata* (Guangdong). The phylogenetic relationship among other suborder/superfamily of Decapoda was similar to Ma et al. [36] research.

## Conclusion

In this study, we successfully obtained the mitogenome sequence of the *L. vittata*, which was also the first species of the Hippolytidae to publish the mitogenome sequence in the GenBank database. The genome sequence was 22003 base pairs (bp) and it included 37 genes and three CRs. Each PCGs was initiated by a canonical ATN codon, except for *cox1*, *nad4L* and *cox3*, which were initiated by a TTG, TTG and GTG. Two of the thirteen PCGs (*nad5* and *nad4*) terminated with incomplete stop codon T, and one (*cox1*) terminated with stop codon TAG. The AT-skew (-0.04) and the GC-skew (-0.17) were both negative in the mitogenomes of *L. vittata*. Compared with the gene order of a Decapoda ancestor, the gene arrangement order of the *L. vittata* has changed. Futhermore, phylogenetic analyses showed that *L. vittata* formed a monophyletic branch with other species of the genus *Lysmata/Exhippolysmata*.

## Supporting information

**S1 Fig. Comparison of the difference interval between *L. vittata* (Fujian) and *L. vittata* (Guangdong) mitogenome sequence.**
(TIF)

**S2 Fig. Predicted secondary structure for the tRNAs of *Lysmata vittata* mitogenome.**
(TIF)

**S1 Table. List of species used to construct the phylogenetic tree.**
(DOC)

**S2 Table. Summary of *Lysmata vittata* mitogenome.**
(DOC)

## Acknowledgments

We thank all laboratory members for their constructive suggestions and discussion.

## Author Contributions

**Funding acquisition:** Zhihuang Zhu, Jianxin Wang, Qi Lin.

**Methodology:** Longqiang Zhu, Leiyu Zhu, Dingquan Wang.

**Software:** Longqiang Zhu.

**Supervision:** Zhihuang Zhu, Leiyu Zhu, Dingquan Wang, Jianxin Wang, Qi Lin.

**Writing – original draft:** Longqiang Zhu.

**Writing – review & editing:** Zhihuang Zhu, Jianxin Wang, Qi Lin.

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
