## [Decision Letter · Decision Letter 0]

29 Jul 2021

PONE-D-21-22008

The complete mitogenome of  Lysmata vittata (Crustacea: Decapoda: Hippolytidae) and its phylogenetic position in Decapoda

PLOS ONE

Dear Dr. Zhu,

Thank you for submitting your manuscript to PLOS ONE. After careful consideration, we feel that it has merit but does not fully meet PLOS ONE’s publication criteria as it currently stands. Therefore, we invite you to submit a revised version of the manuscript that addresses the points raised during the review process.

We look forward to receiving your revised manuscript.

Kind regards,

Tzen-Yuh Chiang

Academic Editor

PLOS ONE

Journal Requirements:

Reviewers' comments:

Reviewer's Responses to Questions

**Comments to the Author**

1. Is the manuscript technically sound, and do the data support the conclusions?

Reviewer #1: No

Reviewer #2: Yes

Reviewer #3: Yes

2. Has the statistical analysis been performed appropriately and rigorously? 

Reviewer #1: Yes

Reviewer #2: Yes

Reviewer #3: Yes

3. Have the authors made all data underlying the findings in their manuscript fully available?

Reviewer #1: Yes

Reviewer #2: Yes

Reviewer #3: Yes

4. Is the manuscript presented in an intelligible fashion and written in standard English?

Reviewer #1: Yes

Reviewer #2: Yes

Reviewer #3: Yes

5. Review Comments to the Author

Reviewer #1: The manuscript presented the first record of mitochondrial genome of the shrimp Lysmata vittata (Crustacea: Decapoda: Hippolytidae) and used it together with mitogenomes of the close families from the GenBank for reconstruction of phylogenetic tree. The data is valuable but only one mitogenome sequence was included in this manuscript. Nowadays, sequencing and describing a mitogenome is routine work and I can not find the novelty of the manuscript. In the manuscript, authors only described the mitogenome sequence and did not compared its structure with close species. Also, authors stated that phylogeny of Hippolytidae is problematic but the study is unable to solve the existing problem. In addition, even though authors stated that “we are the first to publish the mitochondrial genome sequence of the Hippolytidae species in the GenBank database”, a number of mitogenome sequence of Hippolytidae and Lysmatidae are available in the GenBank, such as Lysmata boggessi (GenBank no. MZ144584), Lysmata debelius (GenBank no. MW691200), Saron marmoratus (GenBank no. MT795210) and Exhippolysmata ensirostris (GenBank no. MK681888). It is surprise that authors did not make comparison between L. vittata mitogenome with these species. Moreover, these species were not included in the phylogenetic tree reconstruction. Because of that, the finding and discussion of the manuscript are poor.

Reviewer #2: This manuscript is written with minor grammatical errors. Please improve the manuscript again. The experimental design is properly set up and the problem statement is well defined. I would like to suggest the authors discuss and make a comparison with the available complete mitogenome of the same species, L. vittata from Guangdong province China. It seems that the size of the mitogenomes of both individuals is different even they were collected not far away from each other as Guangdong and Fujian are next to each other. The author should also include both individuals in the phylogenetic tree and the gene arrangement comparison. Apart from this, I also suggest the authors compare all the genes from the mitogenome of each individual and discuss. From this, authors can determine the individual variation of L. vittata from China. Please also improve the phylogenetic tree figures as the figures are quite blurry.

Reviewer #3: The manuscript is well designed and written, the results are sound and helpful for better understanding the mitogenome of Decapoda and for the phylogenetics. Here I have several concerns as below,

The authors should tell the readers why you carried out this study, is there any problem need to be solved in the Background section.

The authors analyzed the phylogenetic relationship under the order Decapoda, however, they introduce limited information in the Introduction section, I suggest the authors to add some information about crabs.

At line 65, what do “fresh leaves” mean?

I wonder for the phylogenetic analysis, did the authors used crabs? if not, they could not use Decapoda.

6. PLOS authors have the option to publish the peer review history of their article (what does this mean?). If published, this will include your full peer review and any attached files.

Reviewer #1: No

Reviewer #2: No

Reviewer #3: **Yes: **Hongyu Ma

---

## [Author Response · Author response to Decision Letter 0]

4 Sep 2021

Reviewer #1: The manuscript presented the first record of mitochondrial genome of the shrimp Lysmata vittata (Crustacea: Decapoda: Hippolytidae) and used it together with mitogenomes of the close families from the GenBank for reconstruction of phylogenetic tree. The data is valuable but only one mitogenome sequence was included in this manuscript. Nowadays, sequencing and describing a mitogenome is routine work and I can not find the novelty of the manuscript. In the manuscript, authors only described the mitogenome sequence and did not compared its structure with close species. Also, authors stated that phylogeny of Hippolytidae is problematic but the study is unable to solve the existing problem. In addition, even though authors stated that “we are the first to publish the mitochondrial genome sequence of the Hippolytidae species in the GenBank database”, a number of mitogenome sequence of Hippolytidae and Lysmatidae are available in the GenBank, such as Lysmata boggessi (GenBank no. MZ144584), Lysmata debelius (GenBank no. MW691200), Saron marmoratus (GenBank no. MT795210) and Exhippolysmata ensirostris (GenBank no. MK681888). It is surprise that authors did not make comparison between L. vittata mitogenome with these species. Moreover, these species were not included in the phylogenetic tree reconstruction. Because of that, the finding and discussion of the manuscript are poor.

Response: Many thanks for your professional suggestions. According to your suggestion, we have added the structural comparison and phylogenetic analysis of L. vittata mitogenome and its relative species in this article. Please see lines 130-143, 214-225 and 240-257. Your suggestions are of great help to us in improving the quality of article.

Reviewer #2: This manuscript is written with minor grammatical errors. Please improve the manuscript again. The experimental design is properly set up and the problem statement is well defined. I would like to suggest the authors discuss and make a comparison with the available complete mitogenome of the same species, L. vittata from Guangdong province China. It seems that the size of the mitogenomes of both individuals is different even they were collected not far away from each other as Guangdong and Fujian are next to each other. The author should also include both individuals in the phylogenetic tree and the gene arrangement comparison. Apart from this, I also suggest the authors compare all the genes from the mitogenome of each individual and discuss. From this, authors can determine the individual variation of L. vittata from China. Please also improve the phylogenetic tree figures as the figures are quite blurry.

Response: Thanks for your suggestions. This manuscript has been revised by an English speaking person. In addition, we have compared and discussed L. vittata (Fujian) and L. vittata (Guangdong) mitogenome sequences and hope that it is now clearer. Please see lines 135-143. Apart from this, we have also improved the phylogenetic tree figures.

Reviewer #3: The manuscript is well designed and written, the results are sound and helpful for better understanding the mitogenome of Decapoda and for the phylogenetics. Here I have several concerns as below,

The authors should tell the readers why you carried out this study, is there any problem need to be solved in the Background section.

Response: Thank you for the suggestion. The purpose and meaning of this research have been added in the introduction section. Please see lines 32-40 and 56-65.

The authors analyzed the phylogenetic relationship under the order Decapoda, however, they introduce limited information in the Introduction section, I suggest the authors to add some information about crabs.

Response: We are grateful for the suggestion. The content of the introduction section has been revised. At present, there are relatively few studies on the complete mitogenomes of Hippolytidae species, so we have selected some species under the order Decapoda to participate in the phylogenetic analysis. The main purpose of our phylogenetic analysis is to understand the classification status of L. vittata in the Hippolytidae, and our analysis results also illustrate the monophyleticity of the genus Lysmata and controversy exists in the classification of some species of Hippolytidae. Therefore, there is no information about crabs in the introduction section, but we added crab information when we performed phylogenetic analysis.

At line 65, what do “fresh leaves” mean?

Response: We apologize for the writing error in the original manuscript. The "fresh leaves" has been revised in the text. Please see line 70.

I wonder for the phylogenetic analysis, did the authors used crabs? if not, they could not use Decapoda.

Response: Thank you for your precious advice. Those advice are all valuable and very helpful for revising and improving our paper. We have added crab data to the phylogenetic analysis content. Please see Supplementary Table 1.

We would like to thank the referee again for taking the time to review our manuscript.

---

## [Decision Letter · Decision Letter 1]

22 Sep 2021

PONE-D-21-22008R1The complete mitogenome of  Lysmata vittata (Crustacea: Decapoda: Hippolytidae) and its phylogenetic position in DecapodaPLOS ONE

Dear Dr. Zhu,

Thank you for submitting your manuscript to PLOS ONE. After careful consideration, we feel that it has merit but does not fully meet PLOS ONE’s publication criteria as it currently stands. Therefore, we invite you to submit a revised version of the manuscript that addresses the points raised during the review process.

We look forward to receiving your revised manuscript.

Kind regards,

Tzen-Yuh Chiang

Academic Editor

PLOS ONE

Reviewers' comments:

Reviewer's Responses to Questions

**Comments to the Author**

1. If the authors have adequately addressed your comments raised in a previous round of review and you feel that this manuscript is now acceptable for publication, you may indicate that here to bypass the “Comments to the Author” section, enter your conflict of interest statement in the “Confidential to Editor” section, and submit your "Accept" recommendation.

Reviewer #1: (No Response)

Reviewer #3: All comments have been addressed

2. Is the manuscript technically sound, and do the data support the conclusions?

Reviewer #1: (No Response)

Reviewer #3: Yes

3. Has the statistical analysis been performed appropriately and rigorously? 

Reviewer #1: Yes

Reviewer #3: Yes

4. Have the authors made all data underlying the findings in their manuscript fully available?

Reviewer #1: Yes

Reviewer #3: Yes

5. Is the manuscript presented in an intelligible fashion and written in standard English?

Reviewer #1: Yes

Reviewer #3: Yes

6. Review Comments to the Author

Reviewer #1: Author’s efforts to improve the submitted manuscript following reviewer’s comments have been done. Therefore, the current version of manuscript is much better than previous one. However, similar to the first comment, the only addition of a single new mitogenome, which does not show special meaning of mtgenome structure, to a crab molecular phylogeny is still problem. In addition, one more crucial problem is that the authors should justify why authors construct the phylogenetic trees for the decapods and what phylogenetic problem authors want to solve in the article. If the authors would like to present a novel decapod tree more comprehensive families and taxonomic representatives should be included. If the authors want to know the phylogenetic position of the Lysmata vittata, the tree will be enough with related species and taxa of L. vittate. In spite of these comments the final decision from the editor charging the article will be respected.

Reviewer #3: The authors have revised the manuscript according my comments, so now I have no more comments. The authors have revised the manuscript according my comments, so now I have no more comments.

7. PLOS authors have the option to publish the peer review history of their article (what does this mean?). If published, this will include your full peer review and any attached files.

Reviewer #1: No

Reviewer #3: **Yes: **Hongyu Ma

---

## [Author Response · Author response to Decision Letter 1]

6 Oct 2021

Reviewer #1: Author’s efforts to improve the submitted manuscript following reviewer’s comments have been done. Therefore, the current version of manuscript is much better than previous one. However, similar to the first comment, the only addition of a single new mitogenome, which does not show special meaning of mtgenome structure, to a crab molecular phylogeny is still problem. In addition, one more crucial problem is that the authors should justify why authors construct the phylogenetic trees for the decapods and what phylogenetic problem authors want to solve in the article. If the authors would like to present a novel decapod tree more comprehensive families and taxonomic representatives should be included. If the authors want to know the phylogenetic position of the Lysmata vittata, the tree will be enough with related species and taxa of L. vittate. In spite of these comments the final decision from the editor charging the article will be respected.

Response: Many thanks for your professional suggestions. We found that the mitogenome structure of L. vittata is different from other species in the same genus. Part of the mitogenome of L. vittata has translocations. There are 3 CR regions, and the length of the genome sequence is about 7 kb longer than that of normal shrimp species. Therefore, this manuscript analyzes the special mitogenome structure of L. vittata. In addition, in the phylogenetic analysis, there are currently limited mitogenome data of Hippolytidae species (all available data was collected), we borrowed the mitogenome data of some species in the Decapoda to conduct phylogenomics and population genetics analysis. Although the dispute over the taxonomic status of Hippolytidae still cannot be resolved, we are making more efforts to resolve this dispute. The purpose of establishing phylogenetic tree is to determine the phylogenetic position of the Lysmata vittata. In order to make this manuscript more express the author's thoughts, we have revised the title of the article, called "The complete mitogenome of Lysmata vittata (Crustacea: Decapoda: Hippolytidae) with implication of phylogenomics and population genetics." Thanks again for your valuable comments on our manuscript.

Reviewer #3: The authors have revised the manuscript according my comments, so now I have no more comments. The authors have revised the manuscript according my comments, so now I have no more comments.

Response: Thank you for taking the time to review our manuscript.

---

## [Decision Letter · Decision Letter 2]

22 Oct 2021

The complete mitogenome of Lysmata vittata (Crustacea: Decapoda: Hippolytidae) with implication of phylogenomics and population genetics

PONE-D-21-22008R2

Dear Dr. Zhu,

We’re pleased to inform you that your manuscript has been judged scientifically suitable for publication and will be formally accepted for publication once it meets all outstanding technical requirements.

Kind regards,

Tzen-Yuh Chiang

Academic Editor

PLOS ONE

Additional Editor Comments (optional):

Reviewers' comments:

Reviewer's Responses to Questions

**Comments to the Author**

1. If the authors have adequately addressed your comments raised in a previous round of review and you feel that this manuscript is now acceptable for publication, you may indicate that here to bypass the “Comments to the Author” section, enter your conflict of interest statement in the “Confidential to Editor” section, and submit your "Accept" recommendation.

Reviewer #1: All comments have been addressed

Reviewer #3: All comments have been addressed

2. Is the manuscript technically sound, and do the data support the conclusions?

Reviewer #1: Yes

Reviewer #3: Yes

3. Has the statistical analysis been performed appropriately and rigorously? 

Reviewer #1: Yes

Reviewer #3: Yes

4. Have the authors made all data underlying the findings in their manuscript fully available?

Reviewer #1: Yes

Reviewer #3: Yes

5. Is the manuscript presented in an intelligible fashion and written in standard English?

Reviewer #1: Yes

Reviewer #3: Yes

6. Review Comments to the Author

Reviewer #1: (No Response)

Reviewer #3: the authors have revised the manuscript according my comments, I think it is suitable to publish it.

7. PLOS authors have the option to publish the peer review history of their article (what does this mean?). If published, this will include your full peer review and any attached files.

Reviewer #1: No

Reviewer #3: **Yes: **Hongyu Ma

---

## [Editor Report · Acceptance letter]

27 Oct 2021

PONE-D-21-22008R2 

The complete mitogenome of *Lysmata vittata* (Crustacea: Decapoda: Hippolytidae) with implication of phylogenomics and population genetics 

Dear Dr. Zhu:

I'm pleased to inform you that your manuscript has been deemed suitable for publication in PLOS ONE. Congratulations! Your manuscript is now with our production department. 

Kind regards, 

on behalf of

Dr. Tzen-Yuh Chiang 

Academic Editor

PLOS ONE